# Effect of Diet and Type of Pregnancy on Transcriptional Expression of Selected Genes in Sheep Mammary Gland

**DOI:** 10.3390/ani9090589

**Published:** 2019-08-21

**Authors:** María Gallardo, Juan G. Cárcamo, Luis Arias-Darraz, Carlos Alvear

**Affiliations:** 1Facultad de Ciencias, Universidad Austral de Chile, PO Box 567, Valdivia 5090000, Chile; 2Centro FONDAP, Interdisciplinary Center for Aquaculture Research (INCAR), Valdivia 5090000, Chile; 3Facultad de Ciencias Veterinarias y Pecuarias, Universidad de Chile, Santa Rosa 11735, Santiago 8320000, Chile

**Keywords:** angiogenesis, lactogenesis, transcriptional expression, lamb muscle

## Abstract

**Simple Summary:**

An experiment was designed to determine the effect of diet and type of pregnancy on the mammary gland development, measured by the transcriptional expression of genes involved in angiogenesis and cell turnover/lactogenesis. To that end, twin and single-bearing ewes were fed naturalized pasture or red clover from day −45 pre-partum until day +60 post-partum, taking samples of mammary tissue at day −10, +30 and +60 post-partum. The results showed that the group of twin-bearing ewes fed red clover was the best combination to increase the expression of genes associated to angiogenesis and cell turnover/lactogenesis in the mammary gland.

**Abstract:**

These trials were carried out to determine firstly the effect of diet and type of pregnancy on the transcriptional expression of genes involved in angiogenesis and cell turnover/lactogenesis inside the sheep mammary gland from late gestation to late lactation. Eighteen Ile de France sheep, 8 twin- and 10 single-bearing ewes were alloted into two groups according to their diet, either based on *ad libitum* naturalized pasture or red clover hay plus lupine from day −45 pre-partum until day +60 post-partum. Samples from diets and mammary glands were collected at day −10 pre partum (time 1), day +30 (time 2) and day +60 post-partum (time 3) and analyzed by qRT-PCR. Additionally, samples from *longissimus dorsi* muscle were taken from lambs twice, at weaning and 45 days later, to determine the effect of the maternal treatment with regard to diet and type of pregnancy, on the mRNA expression of genes involved in lipid metabolism. The data was processed using the lme4 package for R, and SPSS Statistics 23.0 for Windows^®^. The results showed that the group of twin-bearing ewes fed red clover showed a higher expression of genes involved in angiogenesis before lambing and in cell turnover/lactogenesis during late lactation, explained by a lamb survival mechanism to delay apoptosis as a way to keep a secretory cells population and boosted by the diet quality, assuring a longer milk production potential during late lactation. Regarding lambs, apparently the maternal diet would influence the transcriptional expression of lipogenic enzymes in the *longissimus dorsi* muscle after weaning, but further studies are necessary to validate these results. In summary, Twin-bearing ewes fed red clover performed best at increasing the expression of genes associated with angiogenesis and cell turnover/lactogenesis in the mammary gland.

## 1. Introduction

Sheep production is mainly based on pastures and faces serious problems regarding lamb survival and growth from birth until weaning, especially in twin-bearing ewes [1], which is in accordance with the physiological negative energy balance present in peripartum [2,3]. It is known that milk production and its persistence are influenced by the metabolic status of the animals [4].

With regard to the synthesis of milk, the microcirculation in the mammary gland provides the necessary nutrients and oxygen for the lactotrophic mammary cells and also allows for the removal of metabolites [5]. However, there are precedents upon which to state that the processes of angiogenesis and cell proliferation normally occurring in the mamary gland during each lactantion are outdated in time. The mammary parenchyma regulates its own vascularization, which takes place during early lactation [6], unlike that seen in other organs where it occurs in response to an injury [5]. Thus, it is possible to recognize as angiogenic factors certain proteins like VEGF and their receptors VEGFR1 and VEGFR2 (which promote proliferation, survival, migration and differentiation of endothelial cells), ANGPT1 and ANGPT2 (the antagonist protein of ANGPT1, being able to induce endothelial cell apoptosis), and RTK [7]. In addition, some proteins associated to cell turnover/lactogenesis are BAX, BCL2, CCND1, IGF1 and its receptor IGF1R, plus the binding proteins IGFBP3 and IGFBP5, TGFB1 and their receptors TGFB1R1 and TGFB1R2, and LPTR [8].

In the search of forage-based nutritional strategies for peripartum, the lipid supplementation consisting of ω3-polyunsaturated fatty acids (PUFA) turns out to be a good alternative to reduce fat mobilization since it can inhibit the *de novo* synthesis occurring in the mammary gland [9,10,11], and it can also increase the levels of C18:3 ω3, C20:5 ω3, and C22:6 ω3 in the ruminant milk and plasma [12,13,14] with proven benefits for the gestation length and lactation latency [15]. At this point, the 3-year red clover (*Trifolium pratense*, RC), a forage resource resistant to low temperatures and with a high content of 18:3 ω3 PUFA [16], shows some protection against ruminal biohydrogenation [17], by increasing ω3 PUFA content in animal products [18].

Studies on milk production and persistency have determined the effect of parity number on transcriptional expression of ruminant mammary gland; however, there is no data on the effect of type of pregnancy on transcriptional expression mammary gland in ewes. Furthermore, it has been reported that maternal nutrition during pregnancy can predispose to permanent structural, physiological and metabolic changes [19,20]; however, the effect of post-weaning maternal diet on the fetal programming of the offspring is unclear. Therefore, our hypothesis is that both diet and type of pregnancy influence the mRNA expression of genes involved in angiogenesis and cell turnover/lactogenesis in the mammary gland, specifically in ewes fed red clover from late gestation to late lactation. Thus, the objectives of this study were to determine the effect of diet and type of pregnancy on the transcriptional expression of genes involved in angiogenesis and cell turnover/lactogenesis in the mammary gland of twin- and single-bearing ewes fed two diets, and also to report (as descriptive data) the effects on the transcriptional expression of genes involved in the lipid metabolism of lamb muscle after weaning.

## 2. Materials and Methods

### 2.1. Bioethics

The methodology used in this study was approved by the Committee for the Ethical Use of Animals in Experiments of the Universidad Austral de Chile (Nº241/2015).

### 2.2. Location

The experiment was conducted in a farm located 12 km southeast of Villarrica city, IX Region, Chile (39°16′0″ S, 72°13′0″ E), from July 2016 to January 2017.

### 2.3. Animals and Sampling

A random selection of 18 animals was made from a large, free-ranging Ile de France, third-birth sheep flock of similar BCS (3.0), consisting of eight twin- and ten single-bearing ewes, grazing on a naturalized pasture, a successional pasture post cultivation, planted in previous years that germinates spontaneously when the conditions of temperature and humidity are appropriate. This pasture type is dominated by species such as *Agrostis tenuis*, *Holcus lanatus* and *Trifolium repens* [16]. The selected ewes were allocated in two groups according to their diet, either based on *ad libitum* naturalized pasture (NPH, 85.20 ± 1.93% dry matter (DM), 8.80 ± 0.45% crude protein (CP), 2.13 ± 0.07 Mcal kg^−1^ metabolizable energy (ME), 64.77 ± 2.39% neutral detergent fiber (NDF), and 5.26 ± 0.14% total ashes (TA)) or red clover hay (RCH, 83.12 ± 1.62% DM, 10.62 ± 0.12% CP, 2.31 ± 0.04 Mcal kg^−1^ ME, 54.35 ± 0.13% NDF, and 5.23 ± 0.08% TA), supplemented lupine (88.90% DM, 17.17% CP, 3.08 Mcal kg^−1^ ME, 52.72% NDF, and 3.37% TA), according to their requirements [21]. All animals were segregated in four groups, according to the following dietary plan: twin-bearing ewes fed NPH (TP, *n* = 4) or RCH (TC, *n* = 4), and single-bearing ewes fed NPH (SP, *n* = 5) or RCH (SC, *n* = 5). The treatment took place from day from day −45 pre-partum to day +60 post-partum. Samples of mammary gland tissue were obtained at day −10 pre-partum (time 1), day +30 (time 2) and day +60 post-partum (time 3) and analyzed for gene expression by qRT-PCR to obtain DDCT. The biopsy procedure of the mammary gland was according the protocol described by Nielsen et al. [22]. Hence, a pilot study starting at weaning, was conducted taking into account the four above-mentioned maternal groups, where half of lambs of each group was allocated to graze NP and the other half was allocated to graze RC (the twins were assigned to different diets), thus forming 8 groups of lambs. Biopsy samples from *Longissimus dorsi* muscle were taken at weaning and after 45 days of grazing NP and RC, according to the protocol described by Gallardo et al. [16]. Samples for RNA extraction were kept in RNA Safer Stabilizer Reagent (E.Z.N.A.) −80 °C until further analysis at the Institute of Biochemistry and Microbiology, Universidad Austral de Chile.

### 2.4. qRT-PCR Analysis

qRT-PCR analyses for this study were performed with a Lightcycler Mx3005P (Agilent Technologies, CA, USA). The RNA extraction was made from 50 mg of mammary gland and muscle tissue samples was carried out using TRIzol™ Reagent (Invitrogen, ThermoFisher, Waltham, MA, USA), following the procedure indicated by the manufacturer. The final RNA elution was carried out with nuclease-free water. Subsequently, the RNA was quantified by spectrophotometry (maestronano, maestrogen), obtaining the concentration data, in addition the 260/280 and 260/230 ratios were reviewed. Finally, the cDNA preparation was carried out by reverse transcription using 2 ul of RNA in each case. The reverse transcription was carried out using the M-MLV Reverse Transcriptase (Invitrogen, CA, USA). Subsequent qRT-PCR analyses were performed 5 μL of Brilliant II SYBR^®^ Green Master Mix (Agilent Technologies), 0.5 μL forward/reverse (which were designed using Primer-BLAST and further analyzed by Amplifix tool) primer solution (0.2 μmol/L), and 1 μL cDNA template (1:5 dilution ratio) to a thermo cycling program for 10 s at 95 °C, 30 s at 60 °C, and 45 s at 70 °C (45 cycles). Specific oligonucleotides for mammary gland transcripts were designed (Table 1). They were selected from other studies performed in the mammary gland of ruminants during different periods [6,22]. The nucleotides designed for lipid metabolism, such as ACCA (NM_001009256.1), FASN (XM_004013447.1), SCD1 (NM_001009254.1), SREBP1c (XM_004013336.1), and actin (NM_001009784.1) are detailed in Gallardo et al. [23].

Relative mRNA expression was calculated with the comparative efficiency-corrected ΔΔCT method [24]. *β-Actin* gene was stably expressed and served as a reference gene for gene expression normalization in these experiments.

### 2.5. Statistical Analysis

The effects of maternal diet and type of pregnancy on the expression of eight genes related to angiogenesis (*CAIV*, *VEGF*, *VEGFR1*, *VEGFR2*, *ANGPT1*, *ANGPT2*, *MKI67*, and *TBXAS1*) and 13 genes related to cellular turnover/lactogenesis (*LALBA*, *BAX*, *BCL2*, *CCND1*, *IGF1*, *IGF1R*, *IGFBP3*, *IGFBP5*, *LPT*, *LPTR*, *LTF*, *TGFB1*, and *TGFB1R1*) in the mammary gland of ewes were determined by means of a linear model. This study employed fixed factors: a. Diet (to evaluate the effect of diet), b. Type of pregnancy (to determine if the intercepts are different, for diet changes in each type of pregnancy), and c. Diet * Type of Pregnancy interaction (to assess the combined response of diet and pregnancy). The effect of group and measuring time was determined through factorial ANOVA for repeated measures means through SPSS Statistics 23.0 for Windows (IBM Corp, Armonk, NY, USA), and the differences between means were analyzed by the Bonferroni test (*p* < 0.05). The statistical model employed in the analyses consisted of the folllowing: yijk = μ + Ti + Ej + TEij + eijk, where yijk = observation ij; μ = the overall mean; Ti = the effect of diet type i; Ej = the effect of type of pregnancy j; TEij interaction between diet type (i) and type of pregnancy (j), and eijk = random error. *p*-values were estimated with the fold change data.

## 3. Results

### 3.1. Effect on Angiogenesis

The effect of group and measuring time on the transcriptional expression of genes associated to angiogenesis in the mammary gland is shown in Table 2. 

In time 1, group TC showed a higher transcriptional expression for genes *VEGFR2* (together with SP and SC), *ANGPT1*, *ANGPT2*, and *MK167*, when compared to the other groups (*p* < 0.01). In time 2, group TP showed a higher expression level for *CAIV* than group SC (*p* = 0.014). In time 3, higher mRNA expression levels for genes *VEGFR1* and *ANGPT1* (*p* = 0.01) were found in group TC in contrast to the other groups. Regarding time, group SP exhibited a decreased expression for gene *MK167*, and group SC showed decreased levels of *ANGPT2* expression (*p* = 0.04) as the season progressed.

### 3.2. Effect on Cell Turnover/Lactogenesis

The effect of group and measuring time related to the transcriptional expression of genes associated to cell turnover/lactogenesis in the mammary gland is shown in Table 3. In time 1, group TC showed higher expression levels of *BCL2* (*p* = 0.0002), *IGF1* (*p* = 0.0001), and *IGFBP3* (than group TP, *p* = 0.01) as opposed to the other groups. Group SP showed a higher mRNA expression of *TGFB1* than groups TP and TC (*p* = 0.02). In time 2, a higher expression level of gene *LALBA* (*p* = 0.021) was found in groups TP, TC, and SP; group TP showed higher expression levels for genes *BCL2* (*p* = 0.0001), *CCND1* (*p* = 0.007), *IGFBP5* (*p* = 0.008) (together with TC), *IGF1* (*p* = 0.005), and *LPT* (*p* = 0.0002) in contrast to the other groups. During time 3, group TC showed a higher transcriptional expression level for genes *LALBA* (*p* = 0.036), *BAX* (*p* = 0.0001), *BCL2* (*p* < 0.0001), *IGF1* (*p* = 0.0001), *IGFBP3* (*p* = 0.0004), and *IGFBP5* (*p* < 0.0001) when compared with the other groups, together with a higher mRNA expression of gene *CCND1* (*p* = 0.0008) and *IGF1R* (*p* = 0.0001) than that observed in groups SP and SC. Group TP showed a higher expression of *TGFB1* than the other groups (*p* = 0.0002). Regarding time, although group TP displayed an increased expression of *IGF1* (*p* = 0.009) and *TGFB1* (*p* = 0.006) as the season progressed, this group had an increased transcriptional expression of genes *IGFBP3, IGFBP5* (*p* = 0.02), and *LPT* (*p* = 0.009) in time 2, which nonetheless showed a decrease in the last measurement. Although group TC showed a higher mRNA expression of *LALBA* (*p* = 0.023) in times 2 and 3, compared to time 1, and of *IGF1* in times 1 and 3 compared to time 2 (*p* = 0.011), this group had a decreased mRNA expression of *BCL2* (*p* = 0.0017), *CCND1* (*p* = 0.04), *IGF1R* (*p* = 0.027), and *IGFBP5* (*p* = 0.005) in time 2, but increased in time 3. A decrease in the expression of *TGFB1* and *CCND1* was observed in groups SP and SC, respectively, from time 1 to 3.

The effect of maternal diet on the transcriptional expression of genes involved in lipid metabolism of the *longissimus dorsi* muscle of lambs from weaning to 45 days post-weaning (descriptive data, included in the Appendix A
Table A1), showed that the post-weaning maternal diet in grazing lambs would lead to an overexpression of genes encoding for lipogenic enzymes in the *longissimus dorsi* muscle, especially in twins.

## 4. Discussion

The present study was designed to determine the effect of diet and type of pregnancy on the transcriptional expression of genes associated to angiogenesis and cell turnover/lactogenesis in the mammary gland of twin- or single-bearing ewes from late gestation to late lactation. The study also reported descriptive data related to the effect of maternal diet on genes encoding lipid metabolism enzymes present in the intramuscular adipose tissue of lambs born from these ewes, from weaning to 45 days post-weaning.

Unlike to that reported for dairy goats and cows [25,26], the mammary gland development in sheep is mainly completed at lambing with scarce mammary growth during early lactation [27]. Then, cellular metabolic activity and nutrient transport are increased to provide substrates for the synthesis of milk components during early lactation [28], processes mediated by important changes in the transcriptional expression of the mammary gland [29,30]. Also, Colitti and Farinacci [31] reported structural and morphometrical changes associated to cell turnover and transcriptional expression of related genes in sheep mammary glands prior to involution, showing that control of balance between cell growth and death influences the mammary gland development, being very important for a successful lactation.

It is widely known that parity number affects angiogenesis, cell turnover, and survival of cells in the mammary gland during lactation. Safayi et al. [8] reported higher mRNA expression levels of genes involved in angiogenesis, such as *ANGP1*, *ANGPT2*, and *VEGFR2*, and a greater mammary epithelial cell proliferation during early lactation in ≥2 first-parity goats, since the onset of lactation is associated to the udder development and growth during early lactation, processes which are more prolonged in primiparous than in multiparous animals. In the present study, higher mRNA expression levels of genes involved in angiogenesis, such as *ANGPT1, ANGPT2, VEGFR2*, and *MKi67* (a marker of proliferation) were observed in late pregnancy, especially in twin-bearing ewes fed RCH (*p* < 0.05), with a decreasing expression during lactation (*p* < 0.05). The higher expression of angiogenic factors in twin-bearing ewes fed RCH point out to the provision of these factors by the epithelial cells [32], which should follow the same developmental pattern observed in the above-mentioned cells [8], where a greater expression of angiogenic factors should manifest in the group with a lower expression of apoptotic factors; however, it is unknown why this effect, observed in twin-bearing ewes, was not reported in single-bearing ewes.

Regarding cell turnover and lactogenesis, its occurrence has been reported in late lactation [33,34], similar to that seen in the present study, where mRNAs of *IGF1*, *IGF1R*, *IGFBP3*, and *IGFBP5*, plus *BAX*, *BCL2*, and *CCND1* were overexpressed during late lactancy, especially in twin-bearing ewes fed RCH. However, both maternal diets were based on hay in our study; therefore, from these data alone it is difficult to support an explanation to elucidate why the mRNA expression changed in time in both angiogenesis and cell turnover/lactogenesis as a consequence of the diet offered. Thus, the diet becomes relevant when comparing between treatments [35]. Faulconnier et al. [36] determined the effect of extruded linseed alone or in combination with fish oil on the mammary expression of 14 lipogenic genes, five lipogenic enzyme activities, and transcriptomic profiles after slaughtering at day 28. They reported that although extruded linseed alone or combined with fish oil had a significant effect on milk fat content and fatty acid composition, there were no effects on the mRNA expression levels of lipogenic genes; however, at the transcriptome level, they showed more effects on mRNA linked to protein and transport pathways than on lipid metabolism, probably affecting the functional remodeling of the mammary gland, a feature consistent with the findings of Castro-Carrera et al. [37], who fed lactating Assaf ewes with sunflower oil. The animals, slaughtered at day 49, showed that although sunflower oil modified milk fatty acid composition, it had no significant effects on the performance or mRNA expression of lipogenic genes in the mammary tissue, neither on the subcutaneous adipose tissue, suggesting that the effect of diet on the lipid metabolism should be mediated by a post-transcriptional process or by other genes not analyzed by the authors.

In ruminants, although the onset of lactancy is associated to prolactin regulation in the lactotrophic cells of the mammary secretory alveoli, the secretion of milk is related to the expression of LALBA [38], which increases in early lactation since this whey protein forms a key part of the regulatory subunit of the lactose synthase heterodimer, and the beta 1,4-galactosyltransferase constitutes the catalytic component, allowing the enzyme to produce lactose by transfering galactose moieties to glucose during lactancy. Tsiplaukou et al. [39] reported a LALBA under/overexpression following a 70% under or 130% overfeeding of the same diet in lactating dairy sheep, confirming its relationship with both milk protein yield and volume. In the present study, *LALBA* was overexpressed in twin-bearing ewes, especially those fed RCH, however, more studies including milk production measurements are necessary to validate the results.

It has been reported that suckling induced IGF1 release in rats during lactation [40]. However, a higher *IGF1* overexpression in twin-bearing ewes as a consequence of double suckling was not observed during the present study.

Thus, the reason why the treatment only affected the expression of twin-bearing ewes fed RCH, but not that of single-bearing ewes fed the same diet can be explained by the lamb survival mechanism that delays apoptosis as a way to preserve a population of secretory cells to ensure a longer milk production potential in ewes feeding two lambs. At this point, although the pro-apoptotic factors IGFBP5 and BAX were overexpressed in twin-bearing ewes fed RCH, the anti-apoptotic factor BCL2, together with the overexpression of CCND1 (a proliferation-associated protein important in cell cycle regulation) [41], plus IGF1, a mammary mitogen and cell-survival factor [27] and its receptor IGF1R were also overexpressed in the mammary tissue of twin-bearing ewes, especially those fed RC, being able to support the explanation.

It is necessary to mention that in the present study we have included reports on goats and cows to facilitate the analyses of our results, although there are some inter-species differences at the transcriptional level [42], no functionally expressed genes that show antagonistic expression between sheep, goats and cows have been described, suggesting that breastfeeding in ruminants is a conservative process being suitable for comparison purposes [43].

Regarding the transcriptional expression of genes related to lipid metabolism in *Longissimus dorsi* muscle of lambs, it has been determined that a pre-partum maternal diet high in fiber, protein and fat changes the mRNA expression of fetal subcutaneous and perirenal adipose tissues [44]. Gallardo et al. [23] feeding weaned lambs with two different diets, i.e., calafatal and naturalized pasture, reported a higher mRNA expression of *ACCA*, *FAS*, and *SREBF1* in subcutaneous fat of lambs fed calafatal, although the protein expression was not affected by the type of diet. In the present study, the results (analyzed as descriptive data) suggest that the maternal diet influences the transcriptional expression of lipogenic genes in the *longissimus dorsi* muscle after weaning, especially in twin lambs, but further studies are necessary to validate these results.

## 5. Conclusions

Twin-bearing ewes fed red clover hay showed higher expression levels of genes participating in angiogenesis during the dry period and of those genes involved in cell turnover/lactogenesis during late lactation, exhibiting the best combination between diet and type of pregnancy to increase transcriptional expression of genes associated to angiogenesis and cell turnover/lactogenesis in the mammary gland of ewes, thus providing a useful tool to increase profitability of sheep producers.

## Figures and Tables

**Table 1 animals-09-00589-t001:** Primer specifications.

Gene	Accession	Forward Primer Sequence	Amplicon
Angiogenesis		Reverse Primer Sequence	
*CAIV*	XM_012186664.1	F: AGCGCTTTGCCATGGAGATACA	148
		R: AGGGGCTGGAAGTTCACATTCTTG	
*VEGF*	NM_001025110.1	F: TGCTCTACCTTCACCATGCCAA	101
		R: GCGCTGGTAGACATCCATGAACTT	
*VEGFR1 (FTL1)*	XM_015098156.1	F: AGGTGACCTGCTTCAAGCCAAT	106
		R: GAAGGCAGGTGTCGAGTACGTAAA	
*VEGFR2 (KDR)*	NM_001278565.1	F: AAGACGCTGACTTGCTTTGGGA	150
		R: AAATGGGAAGAGCACGCAACCT	
*ANGPT1*	XM_004011787.3	F: GCACCCTCATGCATTCTTGTCA	140
		R: ACCCTTTCCTCTACCCTATCTGCT	
*ANGPT2*	XM_004021671.3	F: GAGACCTGCTCCCAAAGCAGTAAA	145
		R: TCACTGAGTGATGCGGGTTCAA	
*MKI67*	XM_015103501.1	F: TGCAGACTTTGGCACAAACGAC	143
		R: AGTTTTAGCAGGACGCCTGGAA	
*TBXAS1*	XM_012177234.2	F: CATCTTCCTCATTGCTGGCTACGA	143
		R: AGTACTCAGGGGCTGGATGTTTCT	
Cell turnover			
*LALBA*	NM_001009797.1	F: TGCCACCCAGGCTGAACAATTA	106
		R: AAATGCGGTACAGACCCATTCAGG	
*BAX*	XM_015100639.1	F: CTAAGACCTGGTGTAGCCAAGCAA	103
		R: TCGAACCCATGTTCCCTGCATT	
*BCL2*	XM_012103831.2	F: ATGCGGCCCCTGTTTGATTTCT	112
		R: GTGGACTTCACTTATGGCCCAGAT	
*CCND1*	XM_015102997.1	F: ACGACTTCATCGAGCACTTCCTCT	127
		R: GGTGGGTTGGAAATGAACTTCACG	
*IGF1*	XM_012159642.2	F: CCAGACTTTGCACTTCAGAAGC	106
		R: GATGTGACTGGCATCTTCACCT	
*IGF1R*	XM_012098367.2	F: CGAGATCCTGTACATTCGCACCAA	100
		R: GTTCCACTTCACGATCAGCTGAGA	
*IGFBP1*	NM_001145177.1	F: CAGCGATGAGGCCACAGATACAAA	117
		R: CTGGACTCGGTCATCAAGTGGAAA	
*IGFBP3*	NM_001159276.1	F: AGGTTGACTACGAGTCTCAGAGCA	122
		R: CAGGAACTTGAGGTCGTTCAGTGT	
*IGFBP5*	NM_001129733.1	F: TGCGTGGACAAGTATGGGATGAAG	103
		R: AGGGGACGCATCACTCAACATT	
*LPT*	XM_004008038.3	F: ATCCCACTCACCAGCATGCAAA	145
		R: CTACCAAGTGCAAGCACAGTTAGC	
*LPTR*	NM_001009763.1	F: TTGGATGGCCTAGGAATCTGGAGT	105
		R: GTTAGACCCAACCGCTGTCAGAAT	
*LTF*	NM_001024862.1	F: GGTTATTCTGGTGCCTTCAAGTGC	119
		R: AGAAGCTCATACTGGTCCCTGTCA	
*CYP19A*	NM_001123000.1	F: AACACGTCCACATAGCCCAAGT	80
		R: ACCATCTGTGCTGATTCCATCACC	
*TGFB1*	NM_001009400.1	F: GCACGTGGAGCTGTACCAGAAATA	116
		R: GCACAACTCCAGTGACGTCAAA	
*TGFB1R1*	XM_012120354.2	F: CCAAGGAAAACCAGCCATAGCTCA	118
		R: TGTGGCCGAATCATGCCTTACT	
*TGFB1R2*	XM_012099307.2	F: CCTTACAAAGCATGTGGGCTTGAC	132
		R: CCTGCACTGTAGGCGGATTCTTTA	
*ACTIN*	NM_001009784.1	F: TGAAGTGTGACGTGGACATCCGTA	108
		R: AGGTGATCTCCTTCTGCATCCTGT	

CAIV, carbonic anhydrase IV; VEGF, vascular endothelial growth factor A; VEGFR1 (FTL1), fms-related tyrosine kinase 1; VEGFR2 (KDR), kinase insert domain receptor; ANGPT1, angiopoietin 1; ANGPT2, angiopoietin 2; MKI67, marker of proliferation Ki-67; TBXAS1, thromboxane A synthase 1; LALBA, lactalbumin alpha; BAX, BCL2-associated X protein; BCL2: B-cell CLL/lymphoma 2; CCND1, cyclin D1; IGF1, insulin like growth factor 1; IGF1R, insulin like growth factor 1 receptor; IGFBP1, insulin like growth factor binding protein 1; IGFBP3, insulin like growth factor binding protein 3; IGFBP5, insulin like growth factor binding protein 5; LPT, leptin; LPTR, leptin receptor; LTF, lactotransferrin; TGFB1, transforming growth factor beta 1; TGFB1R1, transforming growth factor, beta receptor 1; TGFB1R2, transforming growth factor, beta receptor 2; actin, β-actin.

**Table 2 animals-09-00589-t002:** Effect of group and measuring time (late pregnancy, early and late lactation) related to the transcriptional expression of selected genes (fold changes) associated to angiogenesis in the mammary gland (LSM ± SEM).

Experimental Treatment ^1^	*CAIV*	*VEGF*	*VEGFR1*	*VEGFR2*	*ANGPT1*	*ANGPT2*	*MK167*	*TBXAS1*
Time 1								
TP	1.33 ± 0.35	0.58 ± 0.07	0.41 ± 0.06	0.43 ^b^ ± 0.05	0.31 ^b^± 0.05	0.40 ^b^ ± 0.11	0.36 ^b^ ± 0.12	0.48 ± 0.09
TC	4.81 ± 2.24	1.69 ± 0.63	2.00 ± 0.35	1.89 ^a^ ± 0.18	4.05 ^a^ ± 1.25	2.35 ^a^ ± 0.10	2.11 ^a^ ± 0.28	2.56 ± 0.76
SP	1.56 ± 0.91	1.45 ± 0.23	1.83 ± 0.79	1.39 ^a^ ± 0.17	1.48 ^a,b^ ± 0.47	0.90 ^b^ ± 0.27	1.12 ^a,b^ ± 0.38 ^A^	1.01 ± 0.35
SC	1.77 ± 1.32	1.03 ± 0.18	1.34 ± 0.52	1.15 ^a^ ± 0.30	1.08 ^b^ ± 0.38	1.51 ^a,b^ ± 0.36 ^A^	1.61 ^a^ ± 0.29	1.38 ± 0.40
*p* groups ^2^	0.33	0.17	0.14	0.004	0.01	0.001	0.004	0.06
Time 2								
TP	1.73 ^a^ ± 0.28	1.43 ± 0.39	1.58 ± 0.88	1.03 ± 0.32	1.19 ± 0.48	1.31 ± 0.53	1.02 ± 0.42	1.38 ± 0.45
TC	1.01 ^a,b^ ± 0.20	1.40 ± 0.13	1.37 ± 0.30	1.05 ± 0.04	1.48 ± 0.26	1.34 ± 0.14	1.59 ± 0.49	1.75 ± 0.66
SP	1.52 ^a,b^ ± 0.28	1.20 ± 0.03	1.16 ± 0.13	1.74 ± 0.24	1.98 ± 0.57	1.51 ± 0.27	1.55 ± 0.32 ^A^	1.29 ± 0.21
SC	0.66 ^b^ ± 0.13	0.74 ± 0.17	0.99 ± 0.26	0.85 ± 0.16	0.63 ± 0.13	0.69 ± 0.12 ^B^	2.95 ± 2.13	0.64 ± 0.07
*p* groups	0.014	0.14	0.83	0.07	0.10	0.25	0.76	0.27
Time 3								
TP	1.44 ± 0.45	1.31 ± 0.37	1.04 ^a,b^ ± 0.12	1.00 ± 0.22	0.90 ^b^ ± 0.23	0.96 ± 0.17	2.61 ± 1.04	1.24 ± 0.45
TC	2.28 ± 0.67	1.99 ± 0.59	2.21 ^a^ ± 0.44	1.84 ± 0.37	2.01 ^a^ ± 0.25	2.45 ± 0.69	3.91 ± 1.81	3.13 ± 0.74
SP	0.94 ± 0.15	0.94 ± 0.50	0.78 ^b^ ± 0.24	0.94 ± 0.19	0.89 ^b^ ± 0.44	1.40 ± 0.96	0.22 ± 0.06 ^B^	2.66 ± 2.40
SC	0.80 ± 0.43	0.81 ± 0.08	0.88 ^b^ ± 0.20	0.84 ± 0.18	0.94 ^b^ ± 0.15	0.71 ± 0.11 ^B^	0.94 ± 0.25	0.78 ± 0.25
*p* groups	0.18	0.19	0.01	0.06	0.01	0.10	0.12	0.28
P times ^3^								
TP	0.73	0.17	0.32	0.16	0.18	0.19	0.09	0.23
TC	0.18	0.72	0.23	0.06	0.08	0.15	0.33	0.43
SP	0.68	0.56	0.36	0.08	0.36	0.75	0.04	0.69
SC	0.58	0.38	0.64	0.55	0.43	0.04	0.53	0.17

^1^ TP: twin-bearing ewes fed NP; TC: twin-bearing ewes fed RC; SP: single-bearing ewes fed NP; SC: single-bearing ewes fed RC; TP-TC (*n* =4); SP-SC (*n* = 5); ^2^ Different small letters (^a,b^) denote significant differences between groups within each measuring time at *p* ≤ 0.05; ^3^ Different capital letters (^A,B^) denote significant differences of each group according to time *p* ≤ 0.05.

**Table 3 animals-09-00589-t003:** Effect of group and measuring time (late pregnancy, early and late lactation) related to the transcriptional expression of selected genes (fold changes) associated to cell turnover /lactogenesis in the mammary gland (LSM ± SEM).

Exp. Treat ^1^	*LALBA*	*BAX*	*BCL2*	*CCND1*	*IGF1*	*IGF1R*	*IGFBP3*	*IGFBP5*	*LPT*	*LPTR*	*LTF*	*TGFB1*	*TGFB1R1*
Time 1													
TP	2.74 ± 1.96	1.18 ± 0.30	0.91 ^b^ ± 0.23 ^B^	1.01 ± 0.18	0.71 ^b^ ± 0.12 ^B^	1.24 ± 0.14	0.63 ^b^ ± 0.15 ^B^	1.11 ± 0.07 ^AB^	1.02 ± 0.38 ^B^	1.56 ± 0.55	1.72 ± 1.03	1.04 ^a,b^ ± 0.07 ^B^	1.53 ^a^ ± 0.21
TC	1.18 ± 0.27 ^B^	3.14 ± 0.99	2.43 ^a^ ± 0.11 ^A^	1.38 ± 0.07 ^B^	2.69 ^a^ ± 0.28 ^AB^	1.33 ± 0.56 ^B^	3.11 ^a^ ± 1.04	2.03 ± 0.38 ^B^	2.61 ± 1.04	1.36 ± 0.45	1.74 ± 0.11	0.73 ^b^ ± 0.07	0.97 ^a, b^ ± 0.04
SP	0.40 ± 0.21	0.92 ± 0.42	0.73 ^b^ ± 0.09	0.66 ± 0.10	0.85 ^b^ ± 0.26	1.06 ± 0.39	0.81 ^a,b^ ± 0.12	0.71 ± 0.15	2.19 ± 1.38	1.05 ± 0.27	0.62 ± 0.17	1.47 ^a^ ± 0.14 ^A^	1.02 ^a,b^ ± 0.12
SC	2.38 ± 0.94	0.90 ± 0.46	0.89 ^b^ ± 0.23	1.15 ± 0.21 ^A^	0.85 ^b^ ± 0.18	1.02 ± 0.33	0.86 ^a,b^ ± 0.12	0.87 ± 0.33	1.48 ± 0.92	1.09 ± 0.28	2.39 ± 1.67	1.07 ^a,b^ ± 0.10	0.81 ^b^ ± 0.08
*p* Groups ^2^	0.53	0.06	0.0002	0.08	0.0001	0.92	0.01	0.06	0.63	0.80	0.79	0.002	0.008
Time 2													
TP	1.82 ^a^ ± 0.28	1.35 ± 0.19	1.61 ^a^ ± 0.13 ^A^	1.71 ^a^ ± 0.10	1.88 ^a^ ± 0.35 ^A^	2.02 ^a^ ± 0.24	1.50 ^a,b^ ± 0.33 ^A^	1.94 ^a^ ± 0.38 ^A^	5.86 ^a^ ± 1.10 ^A^	1.72 ± 0.54	3.02 ± 2.07	1.29 ± 0.09 ^AB^	1.51 ± 0.27
TC	1.51 ^a^ ± 0.19 ^AB^	1.37 ± 0.23	1.38 ^a^ ± 0.11 ^B^	1.42 ^a^ ± 0.31 ^B^	1.28 ^a,b^ ± 0.16 ^B^	1.36 ^a,b^ ± 0.23 ^B^	2.00 ^a^ ± 0.43	1.39 ^a^ ± 0.28 ^B^	0.53 ^b^ ± 0.18	1.10 ± 0.17	2.93 ± 1.27	0.97 ± 0.05	1.05 ± 0.07
SP	1.70 ^a^ ± 0.25	0.53 ± 0.14	0.80 ^b^ ± 0.13	0.63 ^b^ ± 0.06	0.73 ^b^ ± 0.15	0.66 ^b^ ± 0.05	0.85 ^b^ ± 0.22	0.87 ^b^ ± 0.13	1.31 ^b^ ± 0.42	0.84 ± 0.12	1.46 ± 1.02	0.89 ± 0.06 ^B^	0.76 ± 0.21
SC	0.77 ^b^ ± 0.20	1.23 ± 0.30	0.66 ^b^ ± 0.09	0.82 ^b^ ± 0.16 ^AB^	0.72 ^b^ ± 0.10	0.82 ^b^ ± 0.24	0.70 ^b^ ± 0.21	0.57 ^b^ ± 0.08	1.41 ^b^ ± 0.38	1.07 ± 0.30	1.63 ± 0.77	0.97 ± 0.15	1.05 ± 0.26
*p* Groups	0.021	0.14	0.0001	0.007	0.005	0.005	0.03	0.008	0.0002	0.38	0.77	0.10	0.23
Time 3													
TP	1.09 ^b^ ± 0.26	1.70 ^b^ ± 0.34	1.26 ^b^ ± 0.13 ^AB^	1.40 ^a,b^ ± 0.27	1.89 ^b^ ± 0.19 ^A^	1.93 ^a,b^ ± 0.44	0.65 ^b^ ± 0.09 ^B^	0.85 ^b^ ± 0.12 ^B^	2.61 ± 0.93 ^AB^	1.33 ± 0.36	1.44 ± 0.38	1.75 ^a^ ± 0.17 ^A^	1.24 ± 0.35
TC	2.36 ^a^ ± 0.28 ^A^	2.89 ^a^ ± 0.37	3.02 ^a^ ± 0.35 ^A^	2.64 ^a^ ± 0.50 ^A^	3.23 ^a^ ± 0.54 ^A^	3.45 ^a^ ± 0.66 ^A^	4.91 ^a^ ± 1.16	4.82 ^a^ ± 0.88 ^A^	1.41 ± 0.35	1.17 ± 0.19	1.47 ± 0.48	0.90 ^b^ ± 0.09	1.03 ± 0.16
SP	1.05 ^b^ ± 0.52	0.66 ^b,c^ ± 0.19	0.82 ^b^ ± 0.36	0.61 ^b^ ± 0.07	0.57 ^b,c^ ± 0.15	0.49 ^b^ ± 0.19	0.60 ^b^ ± 0.04	0.52 ^b^ ± 0.05	2.41 ± 1.26	0.76 ± 0.18	1.63 ± 0.74	0.97 ^b^ ± 0.13 ^B^	1.11 ± 0.34
SC	1.17 ^b^ ± 0.26	0.48 ^c^ ± 0.07	0.46 ^b^ ± 0.04	0.54 ^b^ ± 0.06 ^B^	0.41 ^c^ ± 0.06	0.45 ^b^ ± 0.06	0.64 ^b^ ± 0.11	0.53 ^b^ ± 0.06	0.84 ± 0.15	1.29 ± 0.33	1.06 ± 0.52	0.77 ^b^ ± 0.08	0.97 ± 0.05
*p* Groups	0.036	0.0001	<0.0001	0.0008	0.0001	0.0005	0.0004	<0.0001	0.23	0.60	0.87	0.0002	0.83
P times ^3^													
TP	0.62	0.45	0.048	0.09	0.009	0.18	0.02	0.02	0.009	0.85	0.68	0.006	0.73
TC	0.023	0.14	0.0017	0.04	0.011	0.027	0.13	0.005	0.12	0.83	0.41	0.11	0.87
SP	0.10	0.61	0.96	0.92	0.61	0.31	0.47	0.18	0.75	0.58	0.59	0.02	0.58
SC	0.16	0.28	0.14	0.049	0.06	0.26	0.58	0.43	0.73	0.84	0.69	0.22	0.56

^1^ TP: twin-bearing ewes fed NP; TC: twin-bearing ewes fed RC; SP: single-bearing ewes fed NP; SC: single-bearing ewes fed RC; TP-TC (n =4); SP-SC (n = 5); ^2^ Different small letters (^a,b^) denote significant differences between groups within each measuring time at *p* ≤ 0.05; ^3^ Different capital letters (^A,B^) denote significant differences of each group according to time *p* ≤ 0.05.

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
