# Peer review of "Effect of Diet and Type of Pregnancy on Transcriptional Expression of Selected Genes in Sheep Mammary Gland"

_animals, 2019, doi:10.3390/ani9090589_

Round 1
Reviewer 1 Report
The aim of this article is to analyze how the type of lambing of the sheep (simple vs. twin) and the type of diet (normal hay vs. red clover) affect the transcriptional profile of the mammary gland in three temporal moments: 10 days before lambing, 30 and 60 days after lambing. In order to achieve this objective, animals with simple and twin births (10 vs. 8) distributed in two random groups in the two types of diet have been analyzed. Angiogenesis and lactogenesis genes were chosen as indicators of transcriptional activity.
In my opinion, there are numerous aspects that make the manuscript cannot be published in the current state.
Main concerns:
On the one hand, experimental design does not allow the objectives to be met as clearly as one would expect from a manuscript with such an ambitious title. In my opinion, the choice of 8 genes related to angiogenesis and 16 related to cell turnover and/or lactogenesis does not, in any case, imply an adequate sampling to talk about the change of the transcriptomic profile in the mammary gland. It is only an analysis of 24 genes of the more than 16,000 that are expressed physiologically in the sheep mammary gland (doi: 10.1038/srep18399). The title of the manuscript should, therefore, be revised. Furthermore, at the present time the authors should be aware that there are methodological procedures that would allow to answer this question in a more adequate way, such as the global analysis of the mammary transcriptome by means of RNA-seq and this should be considered, at least when talking about the statistical power and the limitations of the experimental design.
Furthermore, nowhere in the manuscript is it described what criteria have been used to choose these genes and what bias we might expect in the results based on that criterion. On what basis do the authors assume that the type of diet and the type of birth will produce a change in the expression of these genes?
On the other hand, there are a number of insufficiently described technical aspects that make it difficult to judge the validity of the results presented in the manuscript:
The chemical composition of the two diets supplied will be supplied. Is there such a different chemical composition that a detectable change is expected in a small number of animals? This aspect needs to be clarified and could limit the validity of the study regardless of the methodology used to measure the change in the transcriptional profile.
Certain methodological aspects are overlooked that could have great importance in the final results such as the RNA extraction protocols, the quality control and integrity measurements of the mRNA (e.g. (RIN). These aspects could introduce an important bias in the efficiency of PCR. Another point is the number of technical replicas used for each PCR amplification. In addition, the authors should bear in mind that, for the delta-deltaCT method, the use of more than one housekeeping gene is highly recommended due to possible biases in amplification efficiency of amplicons.
The discussion of the manuscript refers to a large number of publications of other ruminants and longitudinal studies carried out in the Ovine species are overlooked, both in normal lactation and in supplementation with polyunsaturated fat. I would recommend to the authors to reorient the discussion looking for how to fit their results with those already known in the same species, leaving the results of other species as a secondary objective of the comparison.
There are other minor aspects such as the interpretation of table 2 which is really complicated for a non-expert in the analysis carried out and it should be explained that they mean terms such as LogLik, Deviance). Also to better understand the p-values it should be explained for how many independent tests the Bonferroni correction corrects in each case.
Author Response
Animals
Editorial Office
Chile, Santiago, 08th June, 2019
REF: Manuscript submission for evaluation in Animals, titled “Effect of diet and type of pregnancy on transcriptional expression in sheep mammary gland”.
Dear Editor,
I’m pleased to enclose the revised manuscript which incorporates all of the reviewers' comments. The changes in the manuscript have been clearly highlighted in yellow to make them easily distinguishable to the editor and reviewers. This cover letter includes the individual responses given to each of the reviewers’ comments.
In addition, I would like to thank you for the opportunity to contribute to your Journal with this manuscript entitled “Effect of diet and type of pregnancy on transcriptional expression in sheep mammary gland” This is an original work not published previously nor under consideration for publication elsewhere.
This is a research line funded by FONDECYT Program, Chile, through grants nos. 3160059 and 1150934, and FONDAP Program, through grant no. 15110027.
We truly appreciate all comments made to our wok, and the changes consider the following:
Reviewer #1:
Open Review
(x) I would not like to sign my review report
( ) I would like to sign my review report
English language and style
( ) Extensive editing of English language and style required
( ) Moderate English changes required
( ) English language and style are fine/minor spell check required
(x) I don't feel qualified to judge about the English language and style
Yes Can be improved Must be improved Not applicable
Does the introduction provide sufficient background and include all relevant references? ( ) ( ) (x) ( )
Is the research design appropriate? ( ) (x) ( ) ( )
Are the methods adequately described? ( ) ( ) (x) ( )
Are the results clearly presented? ( ) (x) ( ) ( )
Are the conclusions supported by the results? ( ) ( ) (x) ( )
Comments and Suggestions for Authors
The aim of this article is to analyze how the type of lambing of the sheep (simple vs. twin) and the type of diet (normal hay vs. red clover) affect the transcriptional profile of the mammary gland in three temporal moments: 10 days before lambing, 30 and 60 days after lambing. In order to achieve this objective, animals with simple and twin births (10 vs. 8) distributed in two random groups in the two types of diet have been analyzed. Angiogenesis and lactogenesis genes were chosen as indicators of transcriptional activity.
In my opinion, there are numerous aspects that make the manuscript cannot be published in the current state.
Main concerns:
1. On the one hand, experimental design does not allow the objectives to be met as clearly as one would expect from a manuscript with such an ambitious title. In my opinion, the choice of 8 genes related to angiogenesis and 16 related to cell turnover and/or lactogenesis does not, in any case, imply an adequate sampling to talk about the change of the transcriptomic profile in the mammary gland. It is only an analysis of 24 genes of the more than 16,000 that are expressed physiologically in the sheep mammary gland (doi: 10.1038/srep18399).
Response. Thank you very much, but our aim was not characterize, but to determine the effect of two factors (type of pregnancy and diet) not considered together in any publication until now, on the transcriptional expression of genes already selected and described its expression in the mammary gland of ruminants during different periods (Agenäs et al., 2010; Nielsen et al., 2010, included in references).
2. The title of the manuscript should, therefore, be revised. Furthermore, at the present time the authors should be aware that there are methodological procedures that would allow to answer this question in a more adequate way, such as the global analysis of the mammary transcriptome by means of RNA-seq and this should be considered, at least when talking about the statistical power and the limitations of the experimental design.
Response. Thank you very much, but our aim was determine the effect of two factors (type of pregnancy and diet) on the transcriptional expression of genes already selected and described in the mammary gland of ruminants during different periods, such as gestation and lactation. Thus, we think the tittle and the methodology used are adequate to this study.
3. Furthermore, nowhere in the manuscript is it described what criteria have been used to choose these genes and what bias we might expect in the results based on that criterion. On what basis do the authors assume that the type of diet and the type of birth will produce a change in the expression of these genes?
Response. The selected genes were chosen from other studies performed in the mammary gland of ruminants during different periods (Agenäs et al., 2010; Nielsen et al., 2010, included in references). In the results we expected a differential expression of the selected genes related to angiogénesis and cell turnover/ lactogenesis, as an effect of the two mentioned variables tested in the this study. It is easy to find studies regarding the effect of these two factors separately, but this is the first study including both factors in a factorial design, being the basis of the FONDECYT grant 3160059.
On the other hand, there are a number of insufficiently described technical aspects that make it difficult to judge the validity of the results presented in the manuscript:
4. The chemical composition of the two diets supplied will be supplied. Is there such a different chemical composition that a detectable change is expected in a small number of animals? This aspect needs to be clarified and could limit the validity of the study regardless of the methodology used to measure the change in the transcriptional profile.
Response. The higher protein and metabolizable energy content in red clover compared to naturalized pasture has been widely tested (doi: 10.4067/S0718-58392011000400011); In fact, this study is part of the FONDECYT grant 3160059, based on different studies to validate the benefits of red clover on sheep performance and sheep products. Regarding this paper, the chemical composition of the diet was already included in another publication (Cárcamo et al., 2019, in references). The number of animals was adequate (doi: 10.1002/ejlt.201400033), especially when considering the inclusion of twins (doi: 10.3168/jds.2009-2422).
5. Certain methodological aspects are overlooked that could have great importance in the final results such as the RNA extraction protocols, the quality control and integrity measurements of the mRNA (e.g. (RIN). These aspects could introduce an important bias in the efficiency of PCR.
Response. The RNA extraction was made from 50 mg of mammary gland and muscle tissue samples was carried out using TRIzol™ Reagent (Invitrogen, ThermoFisher), following the procedure indicated by the manufacturer. The final RNA elution was carried out with nuclease-free water. Subsequently, the RNA was quantified by spectrophotometry (maestronano, maestrogen), obtaining the concentration data, in addition the 260/280 and 260/230 ratios were reviewed. Finally, the cDNA preparation was carried out by reverse transcription using 2 ul of RNA in each case. That was included in the manuscript.
RNA cuantification and cDNA preparation (attached file)
The quality control of the prepared cDNA was carried out through a conventional PCR amplifying a fragment of the gene coding for β-actin (108 bp), as seen in the following image (attached file)
Subsequently, the primers (indicated in Table 1), were standarized by qPCR, highlighting a single pick in the melting curves for each set of primers.
In addition, the primer efficiency is summarized in the image below (attached file)
6. Another point is the number of technical replicas used for each PCR amplification. In addition, the authors should bear in mind that, for the delta-delta CT method, the use of more than one housekeeping gene is highly recommended due to possible biases in amplification efficiency of amplicons.
Response. We agree with you; however, the expression of β-actin was stable enough after being challenged with three other possible reference genes: glyceraldehyde-3-phosphate dehydrogenase (GAPDH), ribosomal protein 9, transcript variant (RPS9) and ubiquitously-expressed, prefoldin-like chaperone (UXT).
7. The discussion of the manuscript refers to a large number of publications of other ruminants and longitudinal studies carried out in the Ovine species are overlooked, both in normal lactation and in supplementation with polyunsaturated fat. I would recommend to the authors to reorient the discussion looking for how to fit their results with those already known in the same species, leaving the results of other species as a secondary objective of the comparison.
Response. Thank you for your suggestion. Taking into account the ω3 PUFA apported by red clover (one of the two diets) and the fact that the genes selected came from different studies carried out in other species of ruminants, it was decided to discuss the results including studies in different species of ruminants, which allowed enriching the discussion.
8. There are other minor aspects such as the interpretation of table 2 which is really complicated for a non-expert in the analysis carried out and it should be explained that they mean terms such as LogLik, Deviance). Also to better understand the p-values it should be explained for how many independent tests the Bonferroni correction corrects in each case.
Response. LogLik is a logarithmic transformation that ensures the normal data distribution. Regarding the Word “Deviance” we meant “Deviation”. Regarding Bonferroni test, the model allows the explanation of these comparisons in each of the variables. However, this table was moved of the manuscript, as a reviewer´s suggestion.
It necessary to mention all sections of the manuscript were improved as reviewer´s suggestion.
This research group wishes to express its appreciation for the opportunity to submit the present manuscript to your journal.
Best regards
Dr María Gallardo
MV, MSc, PhD
E-mail: mugallar@gmail.cl
Phone: +562-22328-1364
Universidad Mayor,
Campus Huechuraba
Santiago
Chile

Reviewer 2 Report
The manuscript Effect of diet and type of pregnancy on transcriptional 2 expression in sheep mammary gland, is a potentially interesting manuscript where the authors dive into prenatal diet and singleton vs twin bearing effects on mammary development in ewes via the window of gene expression. Unfortunately many essential parts of the materials and methods are missing, the description of the statistical analysis is confusing, and the discussion does not adequately discuss the results of the paper in light of other studies and is disorganized. The postnatal gene expression results from the lambs are not reported properly and should probably be fully investigated in a separate manuscript.
There is good potential for the mammary gene expression results presented to be a paper, but a lot more work needs to be performed on the materials and methods, and discussion. Specific comments are in the attached edited manuscript. Good luck.

Author Response
Animals
Editorial Office
Chile, Santiago, 08th June, 2019
REF: Manuscript submission for evaluation in Animals, titled “Effect of diet and type of pregnancy on transcriptional expression in sheep mammary gland”.
Dear Editor,
I’m pleased to enclose the revised manuscript which incorporates all of the reviewers' comments. The changes in the manuscript have been clearly highlighted in yellow to make them easily distinguishable to the editor and reviewers. This cover letter includes the individual responses given to each of the reviewers’ comments.
In addition, I would like to thank you for the opportunity to contribute to your Journal with this manuscript entitled “Effect of diet and type of pregnancy on transcriptional expression in sheep mammary gland” This is an original work not published previously nor under consideration for publication elsewhere.
This is a research line funded by FONDECYT Program, Chile, through grants nos. 3160059 and 1150934, and FONDAP Program, through grant no. 15110027.
We truly appreciate all comments made to our wok, and the changes consider the following:
Reviewer #2:
Open Review
(x) I would not like to sign my review report
( ) I would like to sign my review report
English language and style
( ) Extensive editing of English language and style required
(x ) Moderate English changes required
( ) English language and style are fine/minor spell check required
( ) I don't feel qualified to judge about the English language and style
Yes Can be improved Must be improved Not applicable
Does the introduction provide sufficient background and include all relevant references? ( ) (x) ( ) ( )
Is the research design appropriate? ( ) (x) ( ) ( )
Are the methods adequately described? ( ) ( ) (x) ( )
Are the results clearly presented? ( ) () (x) ( )
Are the conclusions supported by the results? ( ) ( ) (x) ( )
Comments and Suggestions for Authors
The manuscript Effect of diet and type of pregnancy on transcriptional expression in sheep mammary gland, is a potentially interesting manuscript where the authors dive into prenatal diet and singleton vs twin bearing effects on mammary development in ewes via the window of gene expression. Unfortunately many essential parts of the materials and methods are missing, the description of the statistical analysis is confusing, and the discussion does not adequately discuss the results of the paper in light of other studies and is disorganized. The postnatal gene expression results from the lambs are not reported properly and should probably be fully investigated in a separate manuscript.
There is good potential for the mammary gene expression results presented to be a paper, but a lot more work needs to be performed on the materials and methods, and discussion. Specific comments are in the attached edited manuscript. Good luck.
Page 2.
-Line 53. The mammary parenchyma regulates its own vascularization, which takes place during early lactation [6], unlike that seen in other organs where it occurs in response to an injury [5], recognizing as angiogenic factors certain proteins like VEGF and their receptors VEGFR1 and VEGFR2 (which promote proliferation, survival, migration and differentiation of endothelial cells), ANGPT1 and ANGPT2 (the antagonist protein of ANGPT1, being able to induce endothelial cell apoptosis), and RTK [7]. In addition, some proteins associated to cell turnover/lactogenesis are BAX, BCL2, CCND1, IGF1 and its receptor IGF1R, plus the binding proteins IGFBP3 and IGFBP5, TGFB1 and their receptors TGFB1R1 and TGFB1R2, and LPTR [8].
Very long sentence that could be broken up into 2 sentences, plus the order of the words after the second comma makes the message that is trying to be conveyed unclear.
Response. Thank you very much. The sentence was divided in two.
-Line 53. Symbols for genes are italicized, whereas symbols for proteins are not italicized (e.g., IGF1). The formatting of symbols for RNA and complementary DNA (cDNA) usually follows the same conventions as those for gene symbols. If many genes are listed together in a table, it is usually up to the authors’ (or the journal’s) discretion as to whether they should be italicized. Gene names that are written out in full are not italicized (e.g., insulin-like growth factor 1). Genotype designations should be italicized, whereas phenotype designations should not be italicized.
Response. We agree with you. The symbols for genes were italicized.
-Line 53. Although expert readers may be familiar with gene and protein symbols, non-expert readers may not be certain about the particular genes or proteins that are being represented. Therefore, it is good practice to provide the full gene or protein name followed by its symbol in parentheses upon first usage (e.g., huntingtin gene (HTT)), particularly if your article is to be published in a journal with broad readership.
Response. Thank you very much. The full name of the genes were included as footnotes because it was the first time we listed all the genes together.
-Line 58. In addition, some proteins associated to cell turnover/lactogenesis are BAX, BCL2, CCND1, IGF1 and its receptor IGF1R, plus the binding proteins IGFBP3 and IGFBP5, TGFB1 and their receptors TGFB1R1 and TGFB1R2, and LPTR [8].
So what? Are they up-regulated? Or just regulated by the mammary parenchyma?
Response. They would be affected by the treatment. Our hypothesis was the type of pregnancy and the diet would affect the transcriptional expression of genes related to angiogenesis and also those related to cell turnover/ lactogenesis.
Line 61. In the search of forage-based nutritional strategies for peripartum… be more specific, is it improved forage -based nutritional strategies? Or strategies that will fix a certain problem?
Response. In Chile sheep production is mainly based on naturalized pastures, also during the critical periods of sheep production, such as peripartum. Considering the tested benefits of red clover in sheep performance and sheep products (doi: 10.4067/S0718-58392011000400011; doi: 10.1007/s11250-019-01893-3), it would be an alternative to consider and evaluate at the moment to select forages to feed ewes during peripartum.
Line 90. A random selection of 18 animals was made from a large, free Ile de France, third-birth sheep…. Extremely long sentence, break up. Free? what does free refer to? Free ranging?
Response. Thank you very much. The sentence was divided in two. Free refers to free ranging Ile de France sheep (it was included in the text).
Page 3
Line 93. All animals were segregated in four groups, according to the following dietary plan: twin-bearing ewes fed NPH (TP) or RCH (TC), and single-bearing ewes fed NPH (SP) or RCH (SC). Briefly describe what naturalized pasture is?
Response. Naturalized pasture is a successional pasture post cultivation. It is a pasture planted in previous years that germinates when temperature and humidity conditions are suitable and occurs spontaneously. This pasture type is dominated by species such as Agrostis tenuis, Holcus lanatus and Trifolium repens (doi: 10.4067/S0718-583920110004).
Line 94. how many animals were in each group?
Response. That was described as footnote in Table 3: TP and TC (n = 3); SP and SC (n = 5) and it was also included in Materials and Methods.
Line 104. grazing NP (14.97% DM, 22.38% CP, 2.67 Mcal kg-1 ME, 29.21% NDF, and 9.66% TA), and RC (17.79% DM, 31.17% CP, 3.01 Mcal kg-1 ME, 29.40% NDF, and 8.71% TA), These are results. Plus the materials and methods should describe how the forage samples were obtained, and how the nutrient analysis was performed.
Response. You are right, these data was transferred to results. Regarding the methodology of forage sampling and analysis, a reference was included in Materials and Methods. Since it was part of the pilot study in lamb muscle, this pasture data, obtained after weaning, was included in Appendix, together with the results obtained from lamb muscle.
Line 107. (E.Z.N.A.) …. until further analysis.
Response. Thank you. This was changed in the text.
Line 111. The RNA extraction was made from 50 mg of mammary gland and muscle tissue samples, which were later analyzed for quantity/purity, and reverse transcribed with the M-MLV Reverse Transcriptase cDNA Synthesis Kit (Invitrogen, California, CA USA). You must describe the RNA extraction process. Describe how RNA was analyzed for quantity/purity, and report their values.
Response. The RNA extraction was made from 50 mg of mammary gland and muscle tissue samples was carried out using TRIzol™ Reagent (Invitrogen, ThermoFisher), following the procedure indicated by the manufacturer. The final RNA elution was carried out with nuclease-free water. Subsequently, the RNA was quantified by spectrophotometry (maestronano, maestrogen), obtaining the concentration data, in addition the 260/280 and 260/230 ratios were reviewed. Finally, the cDNA preparation was carried out by reverse transcription using 2 ul of RNA in each case. That was included in the manuscript.
RNA cuantification and cDNA preparation (attached file)
The quality control of the prepared cDNA was carried out through a conventional PCR amplifying a fragment of the gene coding for β-actin (108 bp), as seen in the following image (attached file)
Subsequently, the primers (indicated in Table 1), were standarized by qPCR, highlighting a single pick in the melting curves for each set of primers.
In addition, the primer efficiency is summarized in the image below (attached file)
Line 112. and reverse transcribed with the M-MLV Reverse Transcriptase cDNA Synthesis Kit (Invitrogen, California, CA USA). Following which protocol?
Response. The cDNA preparation was carried out as shown in the image below (attached file)
cDNA preparation protocol
Line 117. and 1 μL cDNA template … What was the template concentration in terms of initial RNA in the reverse transcription reaction?
Response. For the preparation of cDNA 2 ug of RNA was used, in each case. It is detailed in the Table of cDNA preparation shown above.
Line 122. β-actin gene was stably expressed and served as a reference gene for gene expression normalization in these experiments. What is your evidence for this? Did you try more than 1 housekeeping gene?
Response. The expression of β-actin was stable enough after being challenged with three other possible reference genes: glyceraldehyde-3-phosphate dehydrogenase (GAPDH), ribosomal protein 9, transcript variant (RPS9) and ubiquitously-expressed, prefoldin-like chaperone (UXT).
Line 124. Gene & Accession. Split gene and accession into 2 columns
Response. Thank you. It was changed in the manuscript
Line 140. b. Diet+Type of pregnancy (to determine if the intercepts are different, for diet changes in each type of pregnancy), and c. Diet*Type of Pregnancy interaction (to assess the combined response of diet and pregnancy). These 2 factors are the same. The model should be y = Diet + Type of pregnancy + Diet*Type of pregnancy.
Response. Thank you. It was a typing mistake. It was changed in the manuscript.
Page 5
Line 145. The effect of group … What specially is the effect of group? What group?
Response. It refers to the comparison between tretaments (in this case, we used the word “groups”) in the same time (the study considered 3 times). It was explained as footnote in Table 2.
Line 146. the measuring time … What are the measuring times?
Response. It refers to the comparison of each treatment (or group) in three measuring times (time 1, time 2 and time 3). It was explained as footnote in Table 2.
Line 148. The statistical model employed in the analyses consisted of the folllowing: yijk = μ + Ti + Ej + TEij + eijk, where yijk = observation ij; μ = the overall mean; Ti = the effect of diet type i; Ej = the effect of type of pregnancy j; TEij interaction between diet type (i) and type of pregnancy (j), and eijk = random error. If this is the model you used, why did you state the model on lines 140-141?
Response. Thank you. It was a mistake. It was changed in the manuscript.
Table 2. Effect of diet, type of pregnancy on the transcriptional expression of genes associated to angiogenesis and cell turnover/ lactogenesis in the mammary gland in ewes. which time point is this? If it is overall I would remove it and focus on the next 2 tables.
Response. We agree with you.The Table 2 was rmoved of the manuscript
Table: type, diet *type. Usually tables like this are constructed by putting means and SEM on one side and p-values on the other.
Response. We agree with you. This analyses was very complicated to obtain and understand, and the data obtained was not relevant.
Table: Deviance: WHAT IS DEVIANCE? SEM?
Response. We meant “deviation”
Page 7
Line 206. The effect of maternal diet on the transcriptional expression of genes involved in lipid metabolism of the Longissimus dorsi muscle of lambs from weaning to 45 days post-weaning (descriptive data, included in Appendix), showed that the post-weaning maternal diet in grazing lambs would lead to an overexpression of genes encoding for lipogenic enzymes in the Longissimus dorsi muscle, especially in twins. Remove this from the paper and publish it separately.
Response. We disagree with you, because it is a pilot study, reporting only LSM + SEM and therefore, it is not possible to publish separately. We think these preliminary results will be important to support other future research related to fetal programming.
Page 10
Line 224. In sheep, the onset of lactancy is coordinated at the transcriptional level by the modulation of multiple genes and pathways [32]. Such as? Otherwise this reference doesn't tell you much.
Response. We agree with you. The reference was added to the previous paragraph, to mantain consistency.
Line 230. Regarding maternal diet, Dai et al. [34] elucidated the posible impacts on transcriptome and proteomics associated to a low-quality diet in lactating cows, not only evidenced through a lower milk production, but also a decreased energy metabolism, protein synthesis, mammary cell growth, and increased protein degradation when compared to a high-quality diet, making forage quality become a relevant factor. Dipasquale et al. [35] reported the anti-inflammatory effects of essential fatty acids, which play a role in inhibiting the transcription of pro-inflammatory cytokines by the upregulation of PPARγ expression in bovine mammary epithelial cells. Also, Faulconnier et al. [36] determined the effect of extruded linseed alone or in combination with fish oil on the mammary expression of 14 lipogenic genes, five lipogenic enzyme activities, and transcriptomic profiles after slaughtering at day 28. They reported that although extruded linseed alone or combined with fish oil had a significant effect on milk fat content and fatty acid composition, there were no effects on the mRNA expression levels of lipogenic genes; however, at the transcriptome level, they showed more effects on mRNA linked to protein and transport pathways than on lipid metabolism, probably affecting the functional remodeling of the mammary gland, a feature consistent with the findings of Castro-Carrera et al. [37], who fed lactating Assaf ewes with sunflower oil. The animals, slaughtered at day 49, showed that although sunflower oil modified milk fatty acid composition, it had no significant effects on the performance or mRNA expression of lipogenic genes in the mammary tissue, neither on the subcutaneous adipose tissue, suggesting that the effect of diet on the lipid metabolism should be mediated by a post-transcriptional process or by other genes not analyzed by the authors. There are a lot of reference discussed before any results are; the main reasons for a discussion of a research paper is to discuss the results of the study in light of other published results... what is happening in this paragraph is exactly the opposite with very little discussion of the study results at all.
Response. We agree with you. The paragraph was reduced and restructured.
Line 269. The chemical composition of pastures becomes relevant when comparing between treatments. So,.. are you going to discuss them?
Response. The diet becomes relevant when comparing between treatments.
Page 11
Line 278. In the present study LALBA was overexpressed in twin-bearing ewes, specially those fed RCH, compared to the other groups.o why do you think that happened in light of the references you described?
Response. Thank you, as the milk production was nor considered in the study, the sentence was changed in the manuscript.
Line 292. In the present study, all ewes came from a free flock grazing on NP; therefore, twin-bearing ewes (all with BCS 3.0) randomly assigned to the NPH diet should have shown a lower lamb live weight than those assigned to the RCH diet. So did they have lower weight? The paragraph started off by talking about IGF-1 expression and ended up at lamb weight with really no connection in between. The discussion needs to be organized much better with clear ideas and discussion subjects defined in the paragraphs.
Response. Our apologies. The paragraph was organized and reduced, since the results did not include parameters of performance and therefore, it was moved of the manuscript.
Regarding your question: So did they have lower weight?That was just a projection of the results considered in another publication and therefore, it must be deleted of this manuscript.
Line 295. Thus, the reason why the treatment only affected the expression of twin-bearing ewes fed RCH, but not that of single-bearing ewes fed the same diet can be explained by the lamb survival mechanism that delays apoptosis as a way to preserve a population of secretory cells to ensure a longer milk production potential in ewes feeding two lambs. At this point, although the pro-apoptotic factors IGFBP5 and BAX were overexpressed in twin-bearing ewes fed RCH, the anti-apoptotic factor BCL2, together with the overexpression of CCND1 (a proliferation-associated protein important in cell cycle regulation) [45], IGF1, a mammary mitogen and cell-survival factor [27] and its receptor IGF1R found in the mammary tissue of twin-bearing ewes, specially those fed RC, would support the explanation. Some ideas are coming together, but still needs to be clearly defined and backed up with references.
Response. The explanation was organized to be consistent with the results.
Line 304. It necessary to mention that in the present study we have included reports on goats and cows to facilitate the analyses of our results, although there are some inter-species differences at the transcriptional level, (Tsiplakou et al. 2009), …… Should be mentioned directly when using a cow reference.
Response. We agree with you, but the main results to be discussed were developed extensively in this section and in this case, the sentence was only used to close the discussion, not to discuss specific differences between ruminant species.
Line 309. Regarding the transcriptional expression of genes related to lipid metabolism in Longissimus dorsi muscle of lambs, it has been determined that a pre-partum maternal diet high in fiber, protein and fat changes the mRNA expression of fetal subcutaneous and perirenal adipose tissues [48]. Gallardo et al. [23] feeding weaned lambs with two different diets, i.e., calafatal and naturalized pasture, reported a higher mRNA expression of ACCA, FAS, and SREBF1 in subcutaneous fat of lambs fed calafatal, although the protein expression was not affected by the type of diet. In the present study, the results (analyzed as descriptive data) suggest that the maternal diet influences the transcriptional expression of lipogenic genes in the Longissimus dorsi muscle after weaning, especially in twin lambs, but further studies are necessary to validate these results. Remove. No formal results are shown.
Response. We disagree with you, because it is a pilot study, reporting only LSM + SEM and therefore, it is not possible to publish separately, being only included in Appendix. We think these preliminary results will be important to support other future research related to fetal programming in sheep.
Line 318. Finally, although it is known that maternal lipid supplementation plays a vital role, as that reported by RCH-fed gestating ewes regarding tolerance to cold and lamb survival [49], the present study did not record any dead lambs before weaning regardless of the diet. Remove unless you discuss post-natal results from offspring. To do this properly you have to add a lot more data and MM.
Response. Thank you very much. This was removed of the manuscript.
It necessary to mention all sections of the manuscript were improved as reviewer´s suggestion.
This research group wishes to express its appreciation for the opportunity to submit the present manuscript to your journal.
Best regards
Dr María Gallardo
MV, MSc, PhD
E-mail: mugallar@gmail.cl
Phone: +562-22328-1364
Universidad Mayor,
Campus Huechuraba
Santiago
Chile

Reviewer 3 Report
The effects of the diets, the type of pregnancy and the interaction between them on the transcriptional expression in sheep mammary glands are studied and the results are shown in this manuscript.
Major point:
Precisely, since each group of animals contains in itself the two aspects related to the type of diet and pregnancy, it is not clear, however, how each factor can individually influence them (as it is shown in table 1). In other words, how do you distinguish between diet and pregnancy influences?
Minor points:
line 46 - please delete "i"
lines 155-156 - please correct this sentence, maybe like this "The effects of diet, type of pregnancy and their interaction.........are shown in Table 2"
line 168 - please correct "anigiogenesis"
line 230 - please correct "the posible"
line 304 - please correct "it necessary" with "It is necessary"
line 306 - please insert the number instead of "Tsiplakou et al., 2009"
Author Response
Animals
Editorial Office
Chile, Santiago, 08th June, 2019
REF: Manuscript submission for evaluation in Animals, titled “Effect of diet and type of pregnancy on transcriptional expression in sheep mammary gland”.
Dear Editor,
I’m pleased to enclose the revised manuscript which incorporates all of the reviewers' comments. The changes in the manuscript have been clearly highlighted in yellow to make them easily distinguishable to the editor and reviewers. This cover letter includes the individual responses given to each of the reviewers’ comments.
In addition, I would like to thank you for the opportunity to contribute to your Journal with this manuscript entitled “Effect of diet and type of pregnancy on transcriptional expression in sheep mammary gland” This is an original work not published previously nor under consideration for publication elsewhere.
This is a research line funded by FONDECYT Program, Chile, through grants nos. 3160059 and 1150934, and FONDAP Program, through grant no. 15110027.
We truly appreciate all comments made to our wok, and the changes consider the following:
Reviewer #3:
Open Review
(x) I would not like to sign my review report
( ) I would like to sign my review report
English language and style
( ) Extensive editing of English language and style required
( ) Moderate English changes required
( ) English language and style are fine/minor spell check required
(x) I don't feel qualified to judge about the English language and style
Yes Can be improved Must be improved Not applicable
Does the introduction provide sufficient background and include all relevant references? (x) () ( ) ( )
Is the research design appropriate? ( ) (x) ( ) ( )
Are the methods adequately described? ( ) (x) ( ) ( )
Are the results clearly presented? ( ) (x) ( ) ( )
Are the conclusions supported by the results? (x) ( ) ( ) ( )
Comments and Suggestions for Authors
The effects of the diets, the type of pregnancy and the interaction between them on the transcriptional expression in sheep mammary gland are studied and the results are shown in this manuscript.
Major point:
1. Precisely, since each group of animals contains in itself the two aspects related to the type of diet and pregnancy, it is not clear, however, how each factor can individually influence them (as it is shown in table 1). In other words, how do you distinguish between diet and pregnancy influences?
Response. Each of the factors is involved in the model, which determines that the variance of the error is increasingly smaller and is used for statistical comparison between groups. On the other hand, Table 1 only describes the primers used in this mauscript and in the reference studies.
Minor points:
line 46 - please delete "i"
lines 155-156 - please correct this sentence, maybe like this "The effects of diet, type of pregnancy and their interaction.........are shown in Table 2"
line 168 - please correct "anigiogenesis"
line 230 - please correct "the posible"
line 304 - please correct "it necessary" with "It is necessary"
line 306 - please insert the number instead of "Tsiplakou et al., 2009"
Response. Thank you very much. Changes were included in the manuscript.
It should be noticed that the experimental design and methodology proposed for this study has already been validated in another publication (DOI: 10.1007/s11250-019-01893-3), which allows us to validate the results obtained in this study. The results were re-checked by the co-authors and adjusted according to the changes suggested by the reviewers.
It necessary to mention all sections of the manuscript were improved as reviewer´s suggestion.
This research group wishes to express its appreciation for the opportunity to submit the present manuscript to your journal
Best regards
Dr María Gallardo
MV, MSc, PhD
E-mail: mugallar@gmail.cl
Phone: +562-22328-1364
Universidad Mayor,
Campus Huechuraba
Santiago
Round 2
Reviewer 3 Report
I have no further comments for the Authors
Author Response
There is no information to reply.